# Whole Blood as a Sample Matrix in Homogeneous Time-Resolved Assay—Förster Resonance Energy Transfer-Based Antibody Detection

**DOI:** 10.3390/diagnostics14070720

**Published:** 2024-03-29

**Authors:** Annika Lintala, Olli Vapalahti, Arttu Nousiainen, Anu Kantele, Jussi Hepojoki

**Affiliations:** 1Department of Virology, Faculty of Medicine, Medicum, University of Helsinki, 00290 Helsinki, Finland; 2Department of Veterinary Biosciences, Faculty of Veterinary Medicine, University of Helsinki, 00014 Helsinki, Finland; 3Helsinki University Hospital Diagnostic Center, 00029 Helsinki, Finland; 4Human Microbiome Research Program, Faculty of Medicine, University of Helsinki, 00014 Helsinki, Finland; 5Meilahti Infectious Diseases and Vaccine Research Center, MeiVac, Department of Infectious Diseases, University of Helsinki and Helsinki University Hospital, 00029 Helsinki, Finland; 6Vetsuisse Faculty, Institute of Veterinary Pathology, University of Zürich, 8057 Zürich, Switzerland

**Keywords:** LFRET, whole blood, SARS-CoV-2, COVID-19, antibody detection

## Abstract

The protein-L-utilizing Förster resonance energy transfer (LFRET) assay enables mix-and-read antibody detection, as demonstrated for sera from patients with, e.g., severe acute respiratory syndrome coronavirus 2 (SARS-CoV-2), Zika virus, and orthohantavirus infections. In this study, we compared paired serum and whole blood (WB) samples of COVID-19 patients and SARS-CoV-2 vaccine recipients. We found that LFRET also detects specific antibodies in WB samples. In 44 serum–WB pairs from patients with laboratory-confirmed COVID-19, LFRET showed a strong correlation between the sample materials. By analyzing 89 additional WB samples, totaling 133 WB samples, we found that LFRET results were moderately correlated with enzyme-linked immunosorbent assay results for samples collected 2 to 14 months after receiving COVID-19 diagnosis. However, the correlation decreased for samples >14 months after receiving a diagnosis. When comparing the WB LFRET results to neutralizing antibody titers, a strong correlation emerged for samples collected 1 to 14 months after receiving a diagnosis. This study also highlights the versatility of LFRET in detecting antibodies directly from WB samples and suggests that it could be employed for rapidly assessing antibody responses to infectious agents or vaccines.

## 1. Introduction

We described a wash-free immunoassay utilizing protein L and time-resolved Förster resonance energy transfer (TR-FRET), referred to as the LFRET assay, almost a decade ago [1]. This method represented an improvement in a TR-FRET-based antibody detection assay based on labelled antigens only [2]. LFRET relies on the simultaneous binding of a donor fluorophore-labelled antigen and an acceptor fluorophore-labelled protein L to an antibody molecule. This allows the complex to be detected by TR-FRET [1]. The assay used for measuring antibodies in various sample materials is schematically presented in Figure 1A. We demonstrated that LFRET functions in the serodiagnosis of not only viral infections [3,4,5] but also of autoimmune diseases such as celiac disease [6]. The LFRET assay is widely applicable for antibody detection due to the binding of protein L to the kappa light chain, enabling it to bind to all immunoglobulin classes irrespective of the heavy chain [7]. In the assay, the antigen labelled with the donor fluorophore (e.g., chelated Eu) is brought into close proximity with acceptor fluorophore (e.g., AlexaFluor 647)-labelled protein L by binding to an antibody molecule specific to the antigen [1,2,8]. The close proximity (<100 Å apart) enables FRET between the fluorophores, and the presence of “FRET-active” complexes indicates the presence of antibodies against the antigen utilized (Figure 1A). The long fluorescence emission half-life of the chelated Eu (and other lanthanides) surpasses the autofluorescence inherent to biological samples, allowing for the minimization of background fluorescence when employing time-resolved measurements, i.e., TR-FRET (Figure 1B).

Severe acute respiratory syndrome coronavirus 2 (SARS-CoV-2) is a betacoronavirus responsible for the coronavirus disease 2019 (COVID-19) pandemic [9,10]. Since the initial outbreak in late 2019 in Wuhan, China, reports have described numerous assays for detecting antibodies against SARS-CoV-2 [4,11,12]. These serodiagnostic assays primarily rely on the detection of antibodies specific either to the SARS-CoV-2 nucleoprotein (N), the spike protein (S), or neutralizing antibodies [13,14,15,16,17,18]. Following the roll-out of vaccines that induce an immune response against SARS-CoV-2 S [19], the detection of S-specific antibodies has become less relevant or obsolete in serological COVID-19 diagnoses. However, as the majority of vaccines do not induce an immune response against the SARS-CoV-2 N [20], the detection of N-specific antibodies has become a useful tool for retrospectively demonstrating exposure to SARS-CoV-2 in the vaccinated population [21,22]. The evidence suggests that the titers against SARS-CoV-2 S following vaccination or COVID-19 have a relatively short half-life when judged by the presence of neutralizing antibodies [23,24,25]. Additionally, the continuous emergence of variants that effectively evade the neutralizing antibodies has underscored the need for the continuous boosting of immune responses through re-vaccination [26]. While the development of novel vaccines or drugs for treating COVID-19 could help to reduce the health burden, rapid serodiagnostic assays for evaluating the need for vaccine boosters could offer a more targeted approach for allocating resources.

We recently established the LFRET assay for detecting SARS-CoV-2 S- and N-specific antibodies from serum samples and observed that the SARS-CoV-2 S LFRET results correlate with the presence of neutralizing antibodies [4]. The characteristics of the LFRET assay, particularly the time-resolved detection assay enabled by the long emission half-life of chelated Eu, led us to speculate that LFRET could serve in the detection of antibodies directly from a very complex biological sample matrix such as whole blood (WB). The COVID-19 pandemic presented us with the opportunity of obtaining paired serum and WB samples from individuals with known SARS-CoV-2 vaccination or infection status. To figure out how the LFRET assay results are affected by the presence of red blood cells, platelets, sugars, proteins, and anticoagulants in WB [27], we examined 44 serum and WB sample pairs in the SARS-CoV-2 S LFRET. By analyzing 89 additional WB samples of COVID-19 patients, we compared the LFRET performance from a WB sample against enzyme-linked immunosorbent assay (ELISA) and traditional or pseudovirus-based neutralization tests completed from a serum sample.

## 2. Materials and Methods

### 2.1. Serum and Whole Blood Samples: Collection and Handling

Serum and whole blood (WB) samples from COVID-19 patients were collected under the research permit HUS/211/2020 and the ethics committee approval HUS/853/2020, both granted by Helsinki University Hospital, Finland. The WB samples for protocol optimization were collected in ethylenediaminetertaacetic acid (EDTA) (BD Vacutainer K2E) and heparin tubes (Vacutest) from COVID-19 patients and volunteers, and the serum samples in VACUETTE^®^ Serum Clot activator tubes. After the first EDTA WB (WB-E) and heparin WB thaw, the samples were aliquoted and stored at −20 °C. The aliquots of serum samples were stored at −20 °C. After setting up the assay parameters with serum–WB pairs collected from vaccinated individuals, we analyzed 50 serum–WB–EDTA pairs of patients (N = 44) with mild COVID-19 (not requiring hospital treatment) and healthy controls (N = 6, NEG). Of the patient samples, 12 were collected after less than 30 days, 16 between 30 to 60 days, and 16 over 60 days after the initial symptoms. A pool of NEG samples served as the control throughout testing. After setting up the LFRET assay conditions and observing that the LFRET assay measured from WB correlated with the LFRET results from serum, we studied 89 additional WB-EDTA samples of COVID-19 patients. In total, we studied 139 WB samples, of which 133 were drawn from COVID-19 patients. Of the COVID-19 patients, 54 had received vaccination(s) prior to sample collection.

### 2.2. LFRET Assay Optimization and Set-Up

AlexaFluor647-labelled protein L (AF-L) and europium chelate-labelled (Eu) SARS-CoV-2 spike protein (Eu-S) were described in [4]. In the assay, AF-L serves for tagging all kappa light chain-bearing immunoglobulins with the acceptor fluorophore (Figure 1A). Eu-S was bound with specific immunoglobulins—the presence of antibodies specific to SARS-CoV-2 spike protein in the sample was found to bring the donor and acceptor fluorophore into proximity, allowing the formation of TR-FRET-active complexes (Figure 1A). We set up the LFRET assay in Tris-buffered saline (TBS, Medicago AB, Uppsala, Sweden), supplemented with 0.2% IgG- and protease-free bovine serum albumin (BSA, Jackson ImmunoResearch, West Grove, PA, USA), and pre-mixed the antigen and protein L in the buffer to create a 2-fold master mix. For the assay, we dispensed 10 µL of the 2× master mix into wells of a 384-well plate (PerkinElmer ProxiPlate™ 384 F Plus, Glen Waverley VIC, Australia) and added 10 µL of the diluted (reaction buffer with 3% of BSA) serum or WB-EDTA samples to the mix by pipetting up and down 3–5 times. After mixing the sample and reagents, the plates were incubated at room temperature in the dark until the results were recorded. We optimized the LFRET protocol by cross-titrating the Eu-S (between 0.25 and 2.5 nM on the plate) and AF-L (between 25 and 77.5 nM on the plate) concentrations using a pool of negative SARS-CoV-2 antibody samples and a pool of positive SARS-CoV-2 antibody samples at on-plate dilutions varying from 1:4 to 1:4000. The cross-titration of reagents and samples identified 1.25 nM Eu-S and 77.5 nM as the ideal reagent concentrations for 1:300-diluted WB-EDTA samples. HIDEX Sense was used to record the results (excitation at 330 nm, donor emission at 616 nm, and acceptor emission at 665 nm) at 10 min intervals up to 60 min, and the first measurement was recorded at 10 min after mixing the sample and reagents. All reactions were run in duplicates.

### 2.3. Statistical Analyses

GraphPad Prism version 9.4.1 for Windows was used for calculating and visualizing outliers, as well as for creating figures.

## 3. Results

### 3.1. LFRET Assay Protocol Optimization for WB Samples

We recently reported LFRET assays for the detection of antibodies against SARS-CoV-2 N and S from serum samples [4]. Initial tests showed the reagent concentrations and sample dilutions utilized for serum samples in the earlier study, which were not ideal for whole blood (WB) samples. The cross-titration of assay components identified 1.25 nM Eu chelate-labelled SARS-CoV-2 spike protein (Eu-S) and 77.5 nM AlexaFluor 647-labeled protein L (AF-L) to be ideal on-plate concentrations for 1:300-diluted WB. Interestingly, increasing the WB concentration (i.e., lower dilutions) resulted in lower TR-FRET signals because of decreased Eu chelate emissions. We monitored TR-FRET signal developments every 10 min up to 60 min.

Based on our experience from earlier studies, repeated freeze–thaw cycles do not markedly affect the LFRET signal obtained from serum samples. However, the freezing of WB leads to cell lysis, which could affect antibody detection via, e.g., the release of cellular proteases or the aggregation of antibodies with cellular debris. Hence, we studied the LFRET assay performance of refrozen WB samples and compared the signal with samples that were thawed only once. The results showed that LFRET signals remain at similar levels after two freeze–thaw cycles (Table 1), suggesting that the LFRET assay can be used to produce reproducible results with samples frozen twice. To allow multiple measurements from each sample, we divided the WB samples in aliquots following the initial thaw and kept the thawed samples on ice when performing the assay.

### 3.2. Optimization of the Incubation Time and Normalization Method for the WB LFRET Assay

After optimizing the assay conditions, we studied the effect of incubation and normalization (assay buffer only [TBS] versus a pool of negative samples [NEG]) on the LFRET signals obtained from either serum or WB. To facilitate the comparison, we measured a panel of 44 serum–WB sample pairs collected from patients with an RT-PCR-based COVID-19 diagnosis. None of the patients were vaccinated prior or after SARS-CoV-2 infection at the time of sample collection. The results showed the normalization that was used against TBS and NEG to produce nearly identical LFRET signals for both the WB (Figure 2A) and the serum samples (Figure 2B). This is indicated by the slope coefficient (b) of the graphs averaging at 1.03 for WB and 1.04 for serum, as well as the correlation (R) value of 0.98. We divided the correlation values as follows: very weak (R < 0.2), weak (0.2 < R < 0.4), medium (0.4 < R < 0.6), strong (0.6 < R < 0.8), and very strong (0.8 < R). By comparing the results recorded at 10 min intervals, we identified that 30 min incubation yields the most similar results with two sample matrices, with the slope coefficient (b) being close to 1 (1.01 for both WB and serum). In the end, we chose to utilize the NEG pool for normalizing the results in the subsequent experiments and analyses.

### 3.3. Serum and WB Produce Similar LFRET Signal Levels

Next, we plotted the LFRET signals obtained from serum against those obtained from WB and normalized the measured values with those obtained from the NEG pool. To allow further evaluation, we divided and color-coded the samples according to the sampling time, i.e., dark purple for <30 d (N = 12), light green for 1–2 months (30 d–60 d) (N = 16), and dark green for >2 months (>60 d) (N = 16) for RT-PCR-based COVID-19 diagnoses. The results showed that both sample matrices produced similar LFRET signals (Figure 3). On average, the serum samples produced TR-FRET signals that were 1.17 times higher than the WB samples. Furthermore, the correlation between the LFRET signals obtained from serum versus WB samples varied based on the sampling time point (correlation values for each time point for each sample group are shown in Figure 3). The results indicated a medium-to-strong correlation (R = 0.59–0.78) between the LFRET signals obtained from serum and WB samples collected <30 d after receiving a COVID-19 diagnosis, while the samples collected 1–2 months or >2 months after the diagnosis showed an even stronger correlation (R = 0.6–0.87 and R = 0.93–0.97, respectively) between the sample matrices (Figure 3). The measurements at different time points showed that the TR-FRET signal continued to increase up to 60 min (Figure 3), as previously demonstrated in Figure 2.

### 3.4. LFRET Results Show a Strong Correlation with Neutralizing Antibodies but Correlate Only Weakly with Anti-Spike IgG ELISA Results

After demonstrating that serum and WB samples produce similar LFRET signals, we performed LFRET with additional WB samples collected from patients between 3 and 20 months after receiving an RT-PCR-based COVID-19 diagnosis. We chose these particular samples to draw comparisons between the WB LFRET results in relation to neutralizing antibody levels and anti-spike IgG ELISA results measured from paired serum samples in earlier studies [4,11,28]. After the additional samples, the total number of COVID-19 patient WB samples reached 133, out of which 54 had received at least one vaccination before sampling, 56 had not received vaccination, and 23 were of unknown vaccination status.

The results showed a weak-to-medium correlation between the LFRET signals and the SARS-CoV-2 S IgG ELISA results when the COVID-19 patient samples were divided based on the sampling time (0–1, 1–6, 6–10, 10–14, and >14 months after infection), as shown in Figure 4A. We found that most of the samples collected >1 month after the RT-PCR-based COVID-19 diagnosis produced higher LFRET signals than those collected <1 month following diagnosis. By analyzing the LFRET signals for samples grouped according to the time post vaccination (1–8, 9–16, and >17 weeks after the last vaccine dose; unknown vaccination status [NA]; and non-vaccinated), we found samples collected from recently vaccinated individuals to produce the highest LFRET signals (Figure 4B).

We then compared the LFRET signals to the neutralization titers obtained from paired serum samples. The comparison showed a strong correlation (0.84–0.96) between the LFRET signals and the neutralization titers for patient samples that were drawn more than a month after the RT-PCR-based COVID-19 diagnosis (Figure 4C). We observed a lower correlation between LFRET signals and neutralization test results for samples sorted based on the last vaccine dose (Figure 4D). The samples collected from individuals who were vaccinated produced the highest LFRET signals and neutralization test results (Figure 4D), and the anti-spike IgG ELISA values demonstrated a similar trend (Figure 4B). A complete set of correlations and R^2^ values from each sample group and time point is presented in Appendix A. For R^2^, we used the same explanation model as the one used for correlation purposes to describe the strength of the results. R^2^ was used here to describe how samples divided by time after infection or vaccination status have different (A) antibody levels or (B) neutralization titers. Finally, we compared the anti-spike IgG ELISA results to neutralization test results and found only a moderate (R = 0.45) correlation between the results.

## 4. Discussion

TR-FRET allows the signal measurement window to be set outside the biological material-borne autofluorescence, as presented in Figure 1. Due to this, we speculated that the LFRET assay [1] might allow mix-and-read antibody measurements directly from whole blood (WB). Due to availability of paired WB and serum samples, the COVID-19 pandemic presented us with the opportunity to test this hypothesis with patient material. Due to the absence of suitable negative control samples because of the high vaccine coverage at the time of conducting the study, our focus was not on establishing a cut-off limit for direct diagnostic use, but rather on assessing the suitability of WB as a sample matrix.

Strikingly, even with the presence of cells and cellular debris, hemoglobin, sugars, proteins, and the anticoagulants in WB samples, the presence of antibodies was also revealed by the LFRET assay. Furthermore, the LFRET signals obtained from WB appeared to be at similar levels to those obtained from the paired serum samples. Our earlier studies employed serum and plasma as a sample material for the LFRET assay [1,2,8], and indeed we found serum to produce stronger LFRET signals than WB in the present study. The use of WB as the sample matrix for a homogeneous immunoassay is even more challenging than using serum due to, e.g., the chromophores, cells, and cellular debris present in blood. The results of this study suggest that the applicability of WB as a sample material for TR-FRET-based homogeneous assays needs to be studied further. However, such studies are challenged by the lack of suitable archival sample panels for many pathogens.

When setting up the WB LFRET assay, we observed the LFRET signal to increase up to an hour after mixing the sample and the reagents. Both WB and serum behaved in a similar fashion in the assay, indicating that the phenomenon relates to the TR-FRET-active complex formation. The use of an assay buffer versus a pool of negative samples in normalization did not markedly alter the LFRET signal levels. When using WB as the sample material, we observed that a dilution of at least 1:300 was required to prevent the loss of the donor fluorophore signal. We speculate that the observed loss of a donor fluorophore signal at higher WB concentrations (i.e., lower dilutions) is due to the anticoagulants (EDTA or heparin). The extended fluorescence emission half-life of the donor relies on chelated Eu, and it seems plausible that higher anticoagulant concentrations would result in the removal of Eu from the chelate and the subsequent loss of fluorescence. Unlike in our earlier study [4], we found the LFRET signals to increase rather steadily throughout the 60 min monitoring period without a clear optimum reaction time. The steady LFRET signal increase observed herein is likely attributable to variations in the protein L and antigen concentrations. We speculated that higher reagent concentrations would lead to faster signal development. However, the use of high reagent concentrations may also increase the background, and therefore we think that each LFRET assay requires reagent condition optimization. The reagent concentrations and signal levels reached were also dependent on the amount of antibodies in the sample. Eventually, even samples with high antibody concentrations were expected to reach a signal plateau. In line with the idea of reagent concentrations affecting the LFRET signal development time, we found that samples with lower antibody concentrations reached their maximum LFRET signal level faster. It is likely that, similar to our earlier study [4], there is an optimal incubation time when trying to distinguish between positive and negative samples. It seems likely that increasing the incubation time would also result in increased LFRET signals in negative samples due to, e.g., the aggregation of reagents or WB samples.

When comparing the assay performance of WB collected in EDTA tubes versus heparin used as an anticoagulant, we observed that the samples collected in EDTA tubes produced higher LFRET signals and yielded more reliable results. Since the majority of samples in this study were collected in EDTA tubes, we did not optimize the protocol for WB samples collected in heparin tubes. Further studies are needed to assess the suitability of WB samples when collected using other types of anticoagulants, as well as to investigate the potential mechanisms underlying the observed differences.

We found a strong correlation (R = 0.92) between the LFRET signals measured from serum to those obtained from WB samples. Comparing the LFRET signals obtained from WB to anti-spike IgG ELISA results and neutralization test results revealed a better correlation between the LFRET and neutralization test results. This finding may at first seem curious, but it could be attributed to the fact that only a single dilution (1:50) was employed for the anti-spike IgG ELISA test, leading to saturated ELISA reads for the majority of samples. Indeed, the low correlation between the ELISA and neutralization test results supports this interpretation. Interestingly, contrary to our earlier studies [3,5], the WB samples collected several months after the RT-PCR-based COVID-19 diagnosis appeared to produce higher LFRET signals. This observation could be explained by including the information on the latest vaccine dose; the samples producing high TR-FRET signals appeared to come from samples collected from individuals vaccinated after experiencing COVID-19. Such samples would likely contain IgA and IgM, both of which, based on our earlier studies, appear to function well in LFRET [3,6,8]. These antibody classes are also known to possess neutralizing capacity against SARS-CoV-2 [29,30,31], thereby providing a potential explanation for the high correlation between neutralization titers and LFRET signals. High neutralizing antibody levels after COVID-19 or SARS-CoV-2 vaccination reflect strong immune responses and are considered to potentially provide immunity against (re)infection [23,32,33]. COVID-19 patients seroconvert approximately two to three weeks from the onset of symptoms, and the seroconversion occurs faster in patients suffering with more severe symptoms [33,34]. In this study, we employed samples collected from patients that had suffered from mild COVID-19, and due to the relatively low number of samples, we did not attempt to study the effect of age and gender on the LFRET results. Both the severity of COVID-19 and the age of the patient are known to have an impact on the magnitude of the antibody response [35]. Also, the antibody response kinetics following infection and vaccination appear to vary [36,37], which provides yet another point of speculation for the observed differences in the LFRET signals between individuals.

Taken together, our results suggest that LFRET can be utilized for measuring antibodies from diluted WB. The strong correlation between LFRET signals and neutralizing antibody titers indicates that the test could potentially be employed for a quick assessment of antibody levels to guide vaccination decisions and indicate the necessity of vaccine boosters. Such testing could be particularly advantageous for the risk groups, such as the elderly, individuals with autoimmune diseases or those with other predisposing health conditions. In a broader context, LFRET-based measurements from WB could facilitate the rapid monitoring of immunity status against various infectious diseases.

## Figures and Tables

**Figure 1 diagnostics-14-00720-f001:**
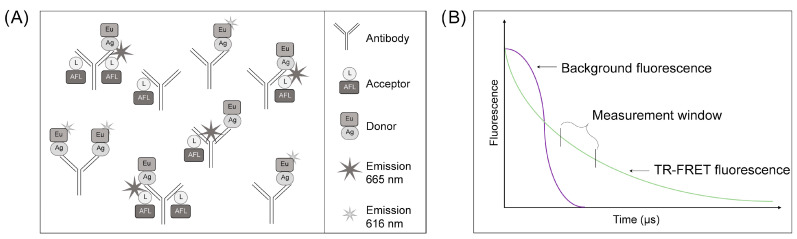
Schematics of the LFRET assay and fluorescence emissions of biological material versus chelated Eu. (**A**) The LFRET assay relies on donor-labelled antigens and acceptor-labelled proteins which are, through simultaneous binding to immunoglobulins specific to the antigen, brought into proximity. When a sample with such complexes is excited at 320–330 nm, FRET occurs, allowing for the detection of acceptor fluorescence at 665 nm. If immunoglobulins/antibodies specific to the antigen are not present in the sample, only donor emissions at 616 nm are detected. (**B**) A schematic illustration of the fluorescence emissions during the TR-FRET measurement process. The optimal window for measuring LFRET signals is located after the short-lived autofluorescence (purple line) from biological material substantially declines but the chelated Eu (green line) signal is still strong.

**Figure 2 diagnostics-14-00720-f002:**
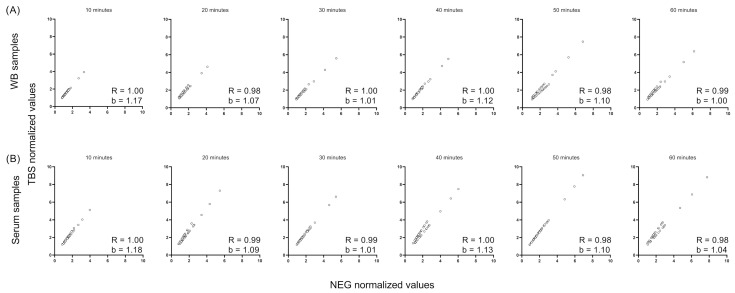
Comparison of the effect of using the buffer (TBS) versus the negative serum pool (NEG) to normalize LFRET signals for (**A**) WB and (**B**) serum samples. Correlation (R) and slope coefficient (b) values were calculated for LFRET values normalized to TBS (values shown on the *Y*-axes) versus LFRET values normalized to the NEG WB/serum sample pool (on the *X*-axes).

**Figure 3 diagnostics-14-00720-f003:**
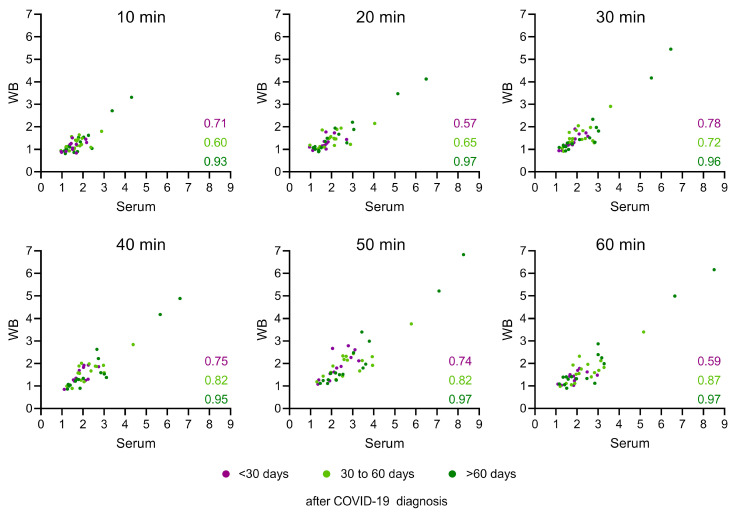
Comparison between WB- and serum-sample-induced LFRET signals. The LFRET signals recorded from serum samples are shown on the *Y*-axis and the corresponding signals from the paired WB samples are shown on the *X*-axis. The samples are color-coded based on the time between the RT-PCR-based COVID-19 diagnosis and sample collection (less than 30 days (dark purple), 30 to 60 days (light green), and more than 60 days (dark green)). The correlation between the WB- and the serum-sample-derived LFRET signals are presented at the bottom right of each graph with a corresponding sample color.

**Figure 4 diagnostics-14-00720-f004:**
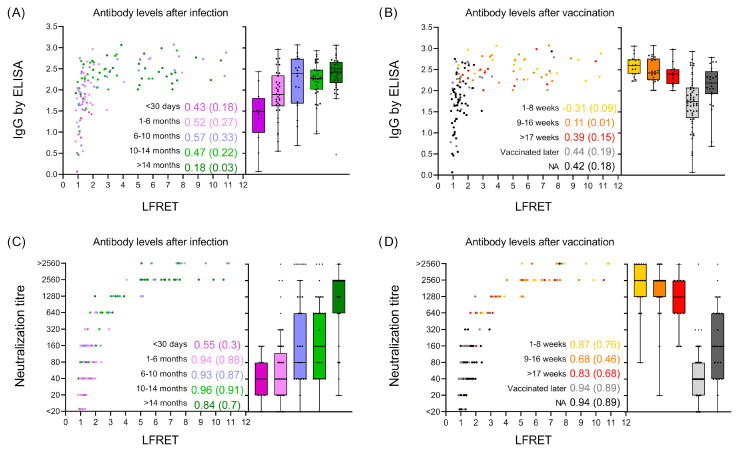
The distribution of anti-spike IgG ELISA OD values and neutralization titers plotted against the WB LFRET results. The LFRET results at 30 min time points are shown in scatter plots on the *X*-axes, and the (**A**,**B**) IgG ELISA OD values and (**C**,**D**) neutralization titers are shown on the *Y*-axes. The samples are color-coded based on the time between the RT-PCR-based COVID-19 diagnosis and sample collection (**A**,**C**) (dark purple, <30 days; light purple, 1–6 months; blue, 6–10 months; light green, 10–14 months; and dark green, over 14 months) and the time since the last vaccine dose (**B**,**D**) (yellow 1–8 weeks; orange, 9–16 weeks; red, more than 17 weeks; light grey, vaccination received after sampling; and dark grey, unknown vaccination status [NA]). The box–whisker plots visualize the distribution of values in each group. The correlation coefficients (R) and R^2^ values in parentheses were calculated for each group and are shown next to the plots.

**Table 1 diagnostics-14-00720-t001:** The effect of freeze–thaw on the LFRET signals measured from WB samples. The table presents the LFRET signals at 20 and 60 min after mixing the components with respect to the assay background for whole blood (WB) samples thawed once or twice (with dilutions of 1:300, 1:400, and 1:600). The results were normalized to a pool of negative (NEG) control samples.

	Dilution of WB	1st Thaw	2nd Thaw
20 min	1:300	1.2	1.7
	1:400	1.2	1.1
	1:600	1.1	1.1
60 min	1:300	2.3	2.3
	1:400	1.7	2.0
	1:600	1.6	1.5

## Data Availability

The raw data of the article will be available upon reasonable request.

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
