# Peer review of "Whole Blood as a Sample Matrix in Homogeneous Time-Resolved Assay—Förster Resonance Energy Transfer-Based Antibody Detection"

_diagnostics, 2024, doi:10.3390/diagnostics14070720_

Round 1

Reviewer 1 Report

Comments and Suggestions for Authors

Although this was hard work, the authors should improve their analyses and discussion concerning the following points:

-       The authors focus on test performance only but effects of age, time-point of seroconversion and immune status regarding neutralizing antibodies of the patients/donors should be discussed.

-        It is intriguing to note, that the assay work with whole blood. Normally, proteomic analysis of blood sera is challenging due to the varying amounts of different proteins. For the detection of antibodies, which are only present in low concentrations, it is necessary to remove other proteins such as albumin. The authors should discuss this.

-       The ethical approval should be given

Author Response

Although this was hard work, the authors should improve their analyses and discussion concerning the following points:

-       The authors focus on test performance only but effects of age, time-point of seroconversion and immune status regarding neutralizing antibodies of the patients/donors should be discussed.

Response: We thank the reviewer for acknowledging the work put into the first version of the manuscript. We agree with the reviewer in that the manuscript is focused primarily on comparing the test performance with serum and blood samples, and that indeed was the main focus of our study (which we also tried to highlight better in the revised introduction). However, the reviewer raises a valid point on the potential factors behind the differences in the observed LFRET signal levels. We have included a bit more discussion around these themes in the discussion section of the revised manuscript. It would indeed be interesting to study a broader sample panel with the test to allow studying the effects of these factors to the LFRET signal and its correlation to neutralizing antibodies.

-        It is intriguing to note, that the assay work with whole blood. Normally, proteomic analysis of blood sera is challenging due to the varying amounts of different proteins. For the detection of antibodies, which are only present in low concentrations, it is necessary to remove other proteins such as albumin. The authors should discuss this.

Response: We do agree with the reviewer, the assay functions surprisingly well given the complexity of the sample material (serum, plasma, and especially whole blood) and the potentially interfering biomolecules present. As encouraged by the reviewer, we have included a few sentences to emphasize this complexity in the discussion section of the revised manuscript.

-       The ethical approval should be given

Response: We thank the reviewer for pointing this out. We have added the ethical approval as a separate section after the manuscript body text, in addition to being mentioned in Materials and Methods section.

Reviewer 2 Report

Comments and Suggestions for Authors

- The Introduction section's paragraph structure should be reversed, starting with the introduction of Sars-cov-2 and moving on to LFRET and its application in antibody detection. The author should clarify why they use of LFRET in the detection of sars-cov-2 antibodies in the final paragraph.

- Provide the Materials section in section 2 Materials and methods. More details about the LFRET assay should also be provided.

- Replace all figures with a high-resolution version. To improve observation, figure 3's light green should be altered to a different hue. 

- The section and subsection titles need to be more accurately modified. 

- Errors in language and paper format must be carefully corrected. For examples, section 2.1 NOT "LFRET assay", maybe "Specimen collection and handling"; "Figure 3. Comparison of LFRET signal obtained from WB to and"; 

Comments on the Quality of English Language

Many language errors must be revised. The authors should recheck all manuscript.

Author Response

- The Introduction section's paragraph structure should be reversed, starting with the introduction of Sars-cov-2 and moving on to LFRET and its application in antibody detection. The author should clarify why they use of LFRET in the detection of sars-cov-2 antibodies in the final paragraph.

Response: We thank the reviewer for the suggestion. We agree in that the manuscript would work in the suggested order of introduction paragraphs, however, our main focus in the study was not to establish another serological test for SARS-CoV-2 but rather to use SARS-CoV-2 as a model to study whether LFRET could be done from whole blood samples. We have speculated this to be the case since the inception of the LFRET assay, however, it required the COVID-19 pandemic for us to get our hands on a panel of paired serum and whole blood samples to allow the comparison. Due to this, we would prefer to keep the order in the introduction as it originally was. For the revised manuscript, we have included a few sentences to the end of introduction to clarify our aims for the study. We hope that the reviewer would agree with us on presenting the introduction in the original order.

- Provide the Materials section in section 2 Materials and methods. More details about the LFRET assay should also be provided.

Response: We thank the reviewer for pointing this out. We have added more detail to the description of the LFRET assay in the revised manuscript.

- Replace all figures with a high-resolution version. To improve observation, figure 3's light green should be altered to a different hue.

Response: We agree with the reviewer in that the resolution of the original figures did not turn out ideal, partially due to format allowed for initial submission. With the revised manuscript, we provide the figures in TIFF format, and hope that the improved figure quality overcomes the coloring issue of the earlier version.

- The section and subsection titles need to be more accurately modified.

Response: We thank the reviewer for pointing this out. We have gone through the manuscript thoroughly and cleaned up the sloppy writing and the errors in the titles.

- Errors in language and paper format must be carefully corrected. For examples, section 2.1 NOT "LFRET assay", maybe "Specimen collection and handling"; "Figure 3. Comparison of LFRET signal obtained from WB to and"; accordingly.

Response: We thank the reviewer for pointing these examples of sloppy writing. For the revised version of the manuscript, we performed overall language polishing in addition to more through revision at some sections of the manuscript. We think that the manuscript improved after these revisions.

Round 2

Reviewer 2 Report

Comments and Suggestions for Authors

I appreciate your effort  to revise thoroughly the manuscript.